

# Total Ozone Dobson, Brewer, Saoz and satellites comparisons at the historical station Arosa

Jean-Pierre Pommereau[1], Florence Goutail[1], René Stübi[2], Geir Braathen[3]

[1] Latmos CNRS Guyancourt, France
[2] MeteoSwiss, Payerne, Switzerland
[3] WMO, Geneva, Switzerland

*Correspondence to:* Jean-Pierre Pommereau (Jean-Pierre.Pommereau@latmos.ipsl.fr)

**Abstract.** Given the importance of the long-term monitoring of the evolution of ozone and following the frequent disruptions of satellites operations, the understanding of quality and stability of ground-based instruments measurements (Dobson, Brewer, SAOZ) performing over several decades, is essential. This is not only for their own records, but also for evaluating the performance of satellites systems and the reprocessing of their data. Many data sets inter-comparisons between Dobson / Brewer networks and satellites are already available (e.g. Redondas et al. 2014 and references therein), for the 40 years of SBUV (Labow et al. 2013) or for the OMI AURA (Balis et al., 2007; McPeters et al. 2008). Here, we evaluate the performance of SAOZ Total Ozone Column (TOC) measurements carried out in Arosa between October 2015 and March 2017 by comparison to simultaneous Dobson, Brewer and satellites observations available there. On average, when using Serdyuchenko et al. (2014) ozone cross-sections for all measurements and correcting the Dobson for its temperature dependence, Dobson, Brewer and SAOZ agree within 1%, providing confidence to the long-term SAOZ measurements carried out all over the world within the international Network for the Detection of Atmospheric Composition Change (NDACC). However, the differences between SAOZ and Dobson/Brewer do show a small seasonality of the order of 1-2% generated by the use of a zonal mean TOMS v8 profile climatology in the SAOZ air mass factor (AMF) calculation, not taking into account seasonal variations.

On the satellites side, the differences with ground based TOCs show larger and highly variable biases (between -1.2% and +2.4%) as well as larger seasonality (around 2-3%) on all satellites, except remarkably with the Solar Backscatter Ultra Violet instrument (SBUV) for which the bias is smaller than 0.4% and the seasonality less than 1%. Asides from ozone absorption cross-sections, most important for satellites TOCs differences are the satellites measurements techniques: ozone profiles for SBUV, in contrast to all other nadir viewing satellites, sensitive like SAOZ, to ozone profile shape assumption.

## 1 Introduction

Following the discovery of the Antarctic Ozone Hole by Farman et al. in 1985, a new instrument, the SAOZ (Système d'Analyse par Observation Zénithale) designed by Pommereau and Goutail (1988), was rapidly deployed at Dumont d'Urville in Antarctica for monitoring the Total Ozone Column (TOC) during the whole year at such a high latitude. Since then, eighteen similar instruments have been built and deployed at all latitudes around the world in the frame of the international NDACC (Network for the Detection of Atmospheric Composition Change).



The SAOZ qualification for NDACC and the long-term stability of the measurements are ensured by
regular blind comparisons with other instruments organised by the NDACC UV-Vis working group
(Hofmann et al., 1995; Vaughan et al., 1997; Roscoe et al., 1999; Vandaele et al. 2005). Following the
aging of the SAOZ components not allowing adequate replacements, a new instrument called Mini-
SAOZ has been designed, first tested in the field in 2009 and now deployed at 12 sites all over the
world. SAOZ and Mini-SAOZ participated in the CINDI 1 and 2 campaigns in 2009 and 2016 in the
Netherlands (Roscoe et al., 2010; Piters et al. 2012, Kreher et al. in preparation). The Mini-SAOZ is a small
UV-Vis spectrometer placed indoor and connected to the outside by an optical fibre. The main
difference with SAOZ is the replacement of the Flat Field Jobin-Yvon spectrometer equipped with a
holographic 360-gr/mm grating and a Hamamatsu 1024 diode array by an Avantes Czerny-Turner
spectrometer equipped with a CCD of 2048 x 16 pixels. One advantage of the Mini-SAOZ is the higher
spectral resolution and the absence of temperature variation of the detector compared to SAOZ,
however, there is an irradiance default of linearity of the new sensor which needs to be corrected.
This default of linearity has been highlighted after the CINDI-2 intercomparison campaign. When
corrected for this default, the Mini-SAOZ fulfils the NDACC requirement criteria and is within 1%
with the SAOZ and the campaign « reference" instrument (Kreher et al. in preparation for
AMT).  Three years of comparisons between the two instruments in the laboratory have shown that, when
the correction is applied, Mini-SAOZ and SAOZ TOCs agree within 2%. The Mini-SAOZ can thus join the
NDACC network, under the unique SAOZ name.
The objective of the present study is to investigate the accuracy of TOC measurements of the Mini-SAOZ
UV-Visible spectrometer compared to older well characterized Dobson and Brewer instruments and thus its
use for long-term ozone monitoring and satellites validation. The study has been performed at the ozone
historical station of Arosa in Switzerland by comparison of coincident daily mean observations available
there. Indeed, aside from those of the Mini-SAOZ deployed between October 2015 and March 2017, other
TOC measurements available at Arosa are those of three ground-based instruments, the DOBSON #62 and
#51 and the BREWER #40 and of several satellites operated by NASA and NOAA and by the European
ESA and EUMETSAT agencies.
The paper is organised as follows. Section 2 provides a description of the Mini-SAOZ and other ground-
based instruments and of the data available from them, which are compared in section 3. Section 4 provides
the details of satellites measurements operating during the same period, which data are compared in section
5. The conclusions are summarized in section 6.

## 32   2 Ground based TOC measurements

The data used in the study are the ground-based Mini-SAOZ UV-Vis zenith sky spectrometer, and those of
Dobson and Brewer direct-sun UV spectrophotometers. The instruments, their performances and their
measurements are described below.

### 36   2.1 Mini-SAOZ



The SAOZ instrument, developed in 1988 is progressively replaced by Mini-SAOZ developed since 2009.
The SAOZ/Mini-SAOZ network is now made of 40 instruments spread all over the globe. They are all UV-
Visible spectrometers observing sunlight scattered by the atmosphere at zenith to retrieve $O_3$ and $NO_2$
columns at sunset and sunrise in all weather conditions (Pommereau and Goutail, 1988). The species are
measured by a method of differential absorption in the visible Chappuis bands between 450-550 nm. Ozone
absorption cross-sections used in the retrieval are those of Bogumil et al. (2003). The Mini-SAOZ is
measuring ozone slant columns. The TOC are calculated using a daily AMF (Air mass factor) based on
TOMS v8 zonal mean ozone profiles climatology (Hendrick et al, 2011). The Mini-SAOZ is the reference
instrument for this study providing the data to which all other measurements will be compared.

### 10    2.2 Dobson UV Spectrophotometer

Designed by Gordon M. B. Dobson at the Oxford University in the mid-1920s, it was first deployed at
Arosa in 1926 for monitoring the ozone column. Asides from its description in the handbook (Dobson, G.
M. B., 1957), a large number of papers relative to its calibration process, performances evaluation
comparisons to other instruments and scientific production, can be found in Redondas et al. 2014 and
references therein. Mainly R&J Beek of London has built about 120 Dobsons, among which about 50
remain in operation today including #51 and #62 at Arosa. The Dobson #62 is providing daily direct sun
TOC measurements based on a double wavelength pairs called AD (A: 305.5 / 325.0 nm and D: 317.5 /
339.9 nm) in the UV in cloud free conditions with ozone slant path (OSP = ozone column x air mass = $O_3$ x
$\mu$) OSP < 1000. However, Dobson #51 is measuring the scattered sunlight at the zenith (called Umkher
measurements) to retrieve the ozone profiles. Since these Umkher data are difficult to compare directly,
they will be ignored in the following. The Dobson ozone column is retrieved using the BP (Bass and Paur,
1985) ozone absorption cross-sections.

### 23    2.3 Brewer UV Spectrophotometer

Designed by Alan W. Brewer in collaboration with T. McElroy and J. B. Kerr at the University of Toronto,
the first Brewer built by Sci-Tec for Environment Canada was installed in Saskatoon in Canada in 1981
(Kerr et al. 2002). Since 2004, Kipp and Zonen in Delft in the Netherlands are building Brewer
instruments. In total, 288 Brewers have been manufactured and deployed all over the world, among which
about 80 are sending their data to the World Ozone and Ultraviolet Radiation Data Centre (WOUDC),
including the Brewer #40 of Arosa. The Brewer is measuring daily direct sun TOC in the UV in cloud free
conditions at OSP $\leq$ ~1500. Their data are analysed by the operational Brewer processor with the BP
absorption cross-sections as those of the Dobson.

### 32    3 Ground based instruments comparisons

Shown on the top of Figure 1 is the comparison between daily mean Dobson #62 (blue), Brewer #40
(black) and Mini-SAOZ (red) TOCs and at the bottom, the percent difference between the two ground-
based direct sun instruments and Mini-SAOZ.
TOC variations of large amplitude are often observed, larger in the winter when the TOC is varying
rapidly, generating noise in the comparisons between non-simultaneous Mini-SAOZ sunrise or sunset and
daytime morning or afternoon direct sun Brewer and Dobson measurements. Also seen on this figure is the
difference of seasonality between the two years, where the TOC remained very low until late December
2015 but varied and increased earlier in October in 2016.  Shown at the bottom of the figure are daily and
monthly mean TOC differences between Mini-SAOZ and others in which outliers larger than 2 standard
deviations have been removed. Compared to Mini-SAOZ, the Dobson shows a mean bias of -0.7 ± 0.4 %
whereas the Brewer is displaying a mean bias of +1.6 ± 0.3% and both are showing a seasonality of
respectively 1.5% and 1.2% amplitude with a March minimum and an August-September maximum, as
summarized in Table 1.

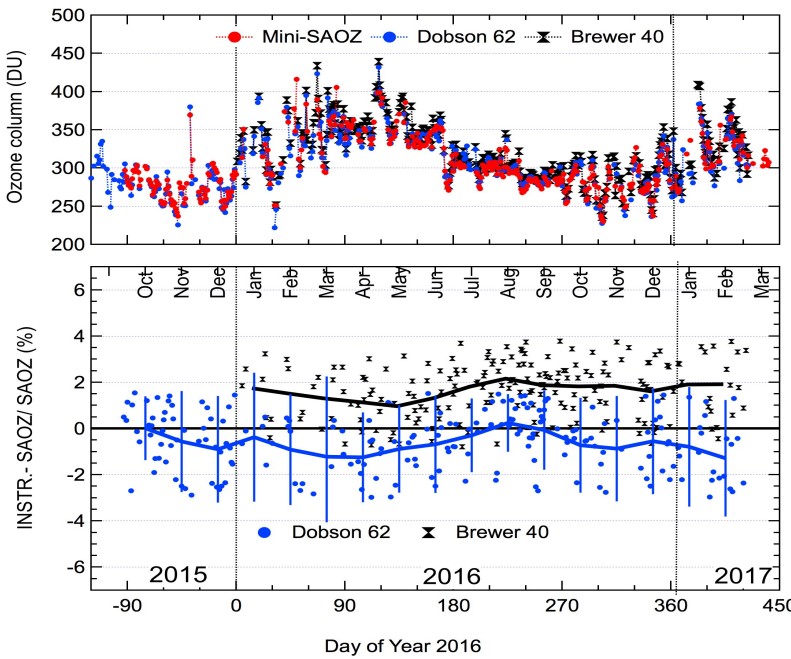

**Figure 1**  Top :  Mini-SAOZ (red), Dobson 62 (blue), and Brewer 40 (black) Total Ozone Column (TOC)
at Arosa between October 2015 and March 2017. Bottom: daily (data points) and monthly mean (solid
lines) percent differences with Mini-SAOZ. The error bars are showing monthly mean standard deviations.
It is larger in March during the period of fast TOC variability.
Part of mean biases between the instruments is coming from the use of different absorption cross-sections,
Bass and Paur (1985) for Dobson and Brewer and Bogumil et al. (2003) for Mini-SAOZ. A correction
corresponding to the use of Serdyuchenko et al., 2014, called SER in the following, has been applied to all
TOC data following the Absorption-Cross-Sections of Ozone (ACSO) WMO-GAW recommendations of
2015 (Orphal et al., 2015). Using SER cross-sections, the Dobson and Mini-SAOZ TOCs increase
respectively by 1% and 0.7%, whereas the Brewer TOC decreases by 0.5%. Shown in Figure 2 are the



biases between Dobson/Brewer and Mini-SAOZ before and after applying the SER correction. On average,
Dobson and Brewer biases are reduced by respectively 0.1% and by 1.3% (Table 1).

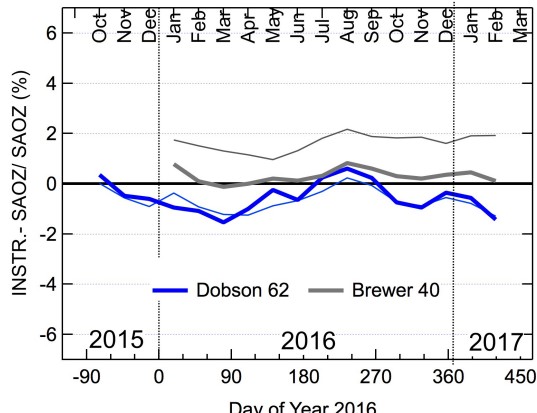

**Figure 2** Relative percent difference between Mini-SAOZ and Dobson 62 (blue) and Brewer (black)
when using SER cross-sections (thick lines) instead of usual operational cross-sections (thin lines).
However, while the seasonality of the difference with the Brewer is smaller than 1%, it is larger (2.1%)
with the Dobson with a spring maximum and fall minimum, but of little different shape on the two years
out of phase of TOC variations seen on the top of Figure 1. Such seasonality is a well-known feature of
Dobson measurements coming from the temperature dependence of the ozone absorption cross-sections
(Redondas et al. 2014). The amplitude of Dobson dependence varies from 0.13%/°C according to Bass and
Paur (1985), to 0.11%/°C from Van Roozendael et al. (1998) and to 0.104%/°C from the most recent
Serdyuchenko et al. (2014) (SER) cross-sections measured by the Institute of Environmental Physics (IUP)
at the University of Bremen.
The temperature seasonality at the altitude of the peak ozone concentration at 30 hPa according to ECMWF
ERA Interim daily simulations would be of the order of 14 °C with a minimum in winter and a maximum
in June-July. The temperature dependence required for correcting the observed TOC differences with Mini-
SAOZ has been modelled and would be of 0.08%/°C for the Dobson. However, a seasonality of the
difference persists, suggesting that something else is responsible for the July-September maximum seen in
Figure 3.
This TOC maximum difference out of phase of temperature modulation as well as part of the inter-annual
variations, can be attributed to the Mini-SAOZ Air Mass Factors calculated from the TOMSv8 zonal mean
profile climatology (McPeters et al. 2007) used to convert measured $O_3$ slant columns into vertical
columns, but not taking well into account the altitude variations of the peak ozone level, for example 0.7
km higher in Aug-Sept 2016 than in Mar 2016 according to Payerne ozone-sondes.





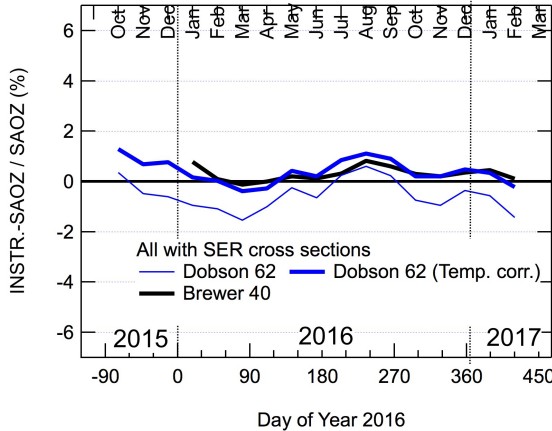

**Figure 3** Relative per-cent difference between Mini-SAOZ and Dobson 62 (blue) before (blue thin line)
and after (blue bold line) correcting the Dobson for a 0.08%/°C temperature dependence.
In conclusion, after applying SER absorption cross-sections to all, and temperature correction on the
Dobson, the differences with Mini-SAOZ show biases <0.5% and a standard deviation <0.5%. However,
Dobson and Brewer are still showing a mean seasonality of respectively 1.7% and 0.9% in the difference
with the Mini-SAOZ, with a spring maximum and fall minimum, but of little different shape on the two
years in phase with TOC variations seen on the top of Figure 1. This seasonality as well as part of the inter-
annual variations is attributed to the Mini-SAOZ Air Mass Factors used to convert $O_3$ slant columns into
vertical columns, not taking well into account the altitude variations of the peak ozone level.
**4 Satellites TOC measurements**
The satellites measurements used in this study are those of the ESA/NASA Ozone Monitoring Instrument
(OMI), the NOAA Solar Backscatter Ultra Violet (SBUV) Nimbus 19 instrument, the NASA SUOMI
National Polar Orbiting Satellite Partnership (SUOMI-NPP) and those of the European EUMESAT Global
Ozone Monitoring Experiment 2-A and 2-B (GOME2-A, GOME2-B) and the Infrared Atmospheric
Sounding Interferometer (IASI). Most of the data used here are those available in real-time on the NASA
website (acd-ext.gsfc.nasa.gov and avdc.gsfc.nasa.gov) except those of IASI A and B available on
https://cds-espri.ipsl.upmc.fr/IASI and those of OMI and GOME 2 retrieved with the GODFITv4 processor
(Lerot et al. 2014, Garane et al., 2018) in the frame of the scientific ESA/CCI-2 project and available on
www.esa-ozone-cci.org.
**4.1 OMI**
The Ozone Monitoring Instrument (OMI) is a nadir viewing spectrometer launched in July 2004 aboard
EOS-Aura in a solar-heliosynchronous polar orbit at 705 km altitude by NASA with a spatial resolution of



25 km x 13 km and an equator crossing time at 13:45 UTC which has provided ozone data since August 9,
2004 (Levelt et al., 2006). Three sets of data will be used: OMI-TOMS using TOMS processing heritage
OMI-DOAS based on KNMI (Royal Netherlands Meteorological Institute) DOAS fitting algorithms,
both using BDM absorption cross-sections and TOMS v8 ozone profile climatology, and a third data set
OMI-GODFIT v4 (Garane et al., 2018) using SER cross-sections and a new ozone profiles climatology
based on ozone-sondes and MLS (Microwave Limb Sounding) satellite data (Labow et al., 2014) .
**4.2 GOME 2**
The Global Ozone Monitoring Experiment-2 (GOME-2) instrument was flown on the MetOp-A satellite
launched in October 2006 in a polar sun-synchronous orbit at an altitude of 817 km crossing the equator at
9:30 UTC in descending orbit. GOME-2 is a nadir sighting spectrometer developed jointly by ESA and
EUMETSAT (Munro et al., 2006; Munro et al., 2016). Global coverage of the sunny part of the atmosphere
can be achieved in one day. The spatial resolution of GOME-2 is 80 x 40 km, four times smaller than that
of GOME (320 x 40 km) and improved polarization and calibration (Munro et al., 2006). It was followed
by GOME-2B launched on 17 September 2012. Two sets of TOC data are available: those of the NASA
website and those retrieved with the GODFIT v4 algorithm developed in the frame of the CCI project
described above.
**4. 3 SBUV**
The Solar Backscatter UltraViolet (SBUV) is a series of 11 satellite instruments launched by NASA and
NOAA since April 1970 on Nimbus 4 followed by Nimbus 7 launched in October 1978 and several others
up to Nimbus 19 still working today. The Limb looking instruments are in polar orbit measuring the solar
spectrum backscattered by the atmosphere in four wavelengths bands in the near ultraviolet (312.5, 317.6,
331.2 and 339.9 nm) (Bhartia et al., 1996) with a local equator crossing time between 8:00 and 16:00 UTC.
The spatial resolution of the instrument is 170 x 170 km. SBUV is providing ozone partial columns (DU) in
21 layers. The algorithm used for processing ozone data is the TOMS version 8.6 (Bhartia et al., 2013;
Kramarova et al., 2013; Labow et al., 2013) including temperature corrections (McPeters et al. 2013) when
needed.. The ozone absorption cross-sections are of those of Brion, Daumont and Malicet, (BDM), 1998.
**4.4 SUOMI–NPP**
The OMPS (Ozone Mapping Profiler Suite) was launched in 2011 aboard the SUOMI- NPP (National Polar
Orbiting Partnership) satellite. The OMPS consists of three sensors, the Nadir Mapper (NM), the Nadir
Profiler (NP) and the Limb Profiler (LP), all measuring scattered solar radiances in overlapping spectral
ranges. The nadir sensors are based on the design heritage of the Total Ozone Mapping Spectrometer
(TOMS) and the Ozone Mapping Instrument (OMI). The data of the first one only are considered in this
work. The OMPS Total Column Nadir Mapper (OMPS TC-NM) is measuring ozone total column in the
300-380 nm range with a 50 x 50 km spatial resolution (Flynn et al., 2014). The TOMS v8 algorithm
(Bhartia et al. 2013) is used for TOCs retrievals. The Ozone absorption cross-sections are those of BDM.



The equator crossing time of OMPS-NPP is 1:30 UTC in ascending node and 10:30 UTC in descending
node allowing daytime daily mapping [npp.gsfc.nasa.gov].

### 4.5 IASI

The near-infrared interferometer (IASI) is a nadir sighting sounder designed by CNES (Centre National
d'Etudes Spatiales) and placed aboard the European MetOp satellite launched in 2006. IASI is crossing the
equator at 9:30 UTC in descending mode and at 21:30 UTC in ascending mode. A new IASI-B instrument
was launched on Metop-B in 2012. It has a spatial resolution of 50 x 50 km. IASI measures the infrared
thermal radiation emitted by the surface and the Earth's atmosphere in the 9.6 µm absorption band
(Clerbaux et al. 2009; Hurtmans et al. 2012). Ozone is retrieved with the FORLI-O3 (Fast Optimal
Retrievals and Layers code, updated recently and validated by comparisons with GOME-2, Dobson,
Brewer, SAOZ, and radio-sondes ozone measurements (Boynard, 2018).

### 5 Satellites TOCs Comparisons

Shown in Figure 4 are the TOC monthly mean differences between all NASA/ESA satellites operational
TOCs and Mini-SAOZ above Arosa between Oct 2015 and Mar 2017. The satellite data include OMI-T
(dotted pink), OMI-D (dotted dark blue), GOME2-A (dotted light blue), GOME2-B (dotted yellow),
SUOMI-NPP (green) and SBUV (grey line). Also shown on the same Figure, are the two almost identical
daytime IASI A and B (dotted green). Because of the frequent masking of satellites, cloudy days are
removed by applying cloud-top or cloud thickness filtering.
Displayed in Table 1, are the biases, standard deviations and seasonality of each of them. The biases vary
from -1.2% to +2.1%, the standard deviation from 0.2% to 0.8% and the seasonality from 0.9% to 3.9%.
Most different compared to all others, are SBUV and OMPS, displaying smaller, -0.3% and -0.4% biases
and in the case of SBUV, a far smaller seasonality of 0.9% compared to 2-3% for all others. The
differences between the satellites are partly coming from their spectral ranges and absorption cross-sections
used in the operational retrievals (BDM for TOMS v8 SBUV, OMI-DOAS, GOME2 and NPP, BP for OMI
and 9.6 micron for IASI). But much more important are the differences between observing technics: i.e.
integration of twenty-one layers ozone partial columns for SBUV, whereas all other satellites TOCs are
derived from nadir viewing observations only. Remarkably, the differences do show larger seasonality than
G-B (Dobson and Brewer) and SBUV and in addition higher variability between satellites (2.1 % up to
3.9 %). The reason for that is the use of Averaging Kernels derived from TOMS v8 zonal mean
climatology (McPeters et al. 2007) where the seasonality can be under- or over-estimated depending on the
longitude and the season, in contrast to SBUV using ozone profiles measured at the TOC location. Note
that part of other satellites seasonality might be coming from the Mini-SAOZ Air Mass Factors using the
same zonal mean profiles climatology, but of limited amplitude as shown by the smaller SBUV, Dobson
and Brewer seasonality.

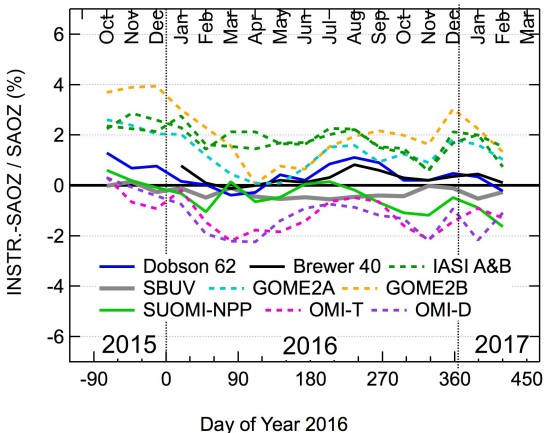

Day of Year 2016

**Figure 4** Total Ozone Column (TOC) differences with Mini-SAOZ: Dobson (dark blue), Brewer (black), GOME2-A (dotted blue), GOME2-B (dotted yellow), OMI-T (dotted pink), OMI-D (dotted violet), SUOMI-NPP (green), IASI A and IASI B (dotted green) and SBUV v8.6 (bold grey).

For testing the influence of differences of cross sections, wavelength ranges and ozone profiles climatology used by the various satellite processors, Mini-SAOZ TOCs have been compared to three satellite data sets all processed with GODFIT v4 using SER cross-sections and the same wavelength ranges. The differences are shown on figure 5: GOME 2-A (light blue), GOME 2–B (yellow) and OMI (pink). In GODFIT v4, the ozone profiles used are a combination of OMI MLS and radio-sondes profiles (Labow at al. 2014). The differences between them are very similar, displaying mean biases of 1.7±1.1%, 2.4±0.8% and 1.9±0.7%, respectively (Table 1). Asides from biases, the differences also show a systematic seasonality of respectively 3.3%, 2.4% and 2.0% amplitude, very similar to that of all other satellites and thus of same origin.

The biases between Mini-SAOZ and all satellites TOCs are smaller than 2.5%. The difference between Mini-SAOZ and operational data sets varying between -1.2% and +2.1%, can be attributed to the cross-sections and wavelength ranges used for each satellite. When using the same GODFIT v4 processing with same SER cross-sections and same wavelength ranges the bias scatter is reduced, varying from +1.7% to 2.4% only (Table 1).

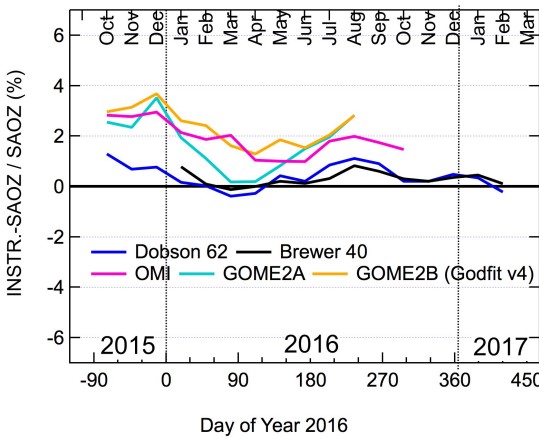

**Figure 5** Total Ozone Column (TOC) differences with Mini-SAOZ: Dobson (dark blue), Brewer (black),
and GODFIT v4 processed GOME 2 A (light blue), GOME 2 B (yellow), OMI (pink).

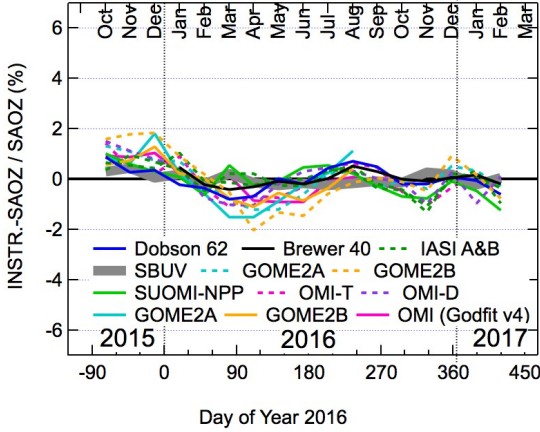

**Figure 6** Same as Figure 4 and Figure 5 but after removing biases.
Figure 6 shows the TOC's differences with Mini-SAOZ after removing the biases. Most of them do show
similar modulations with Nov-Dec 2015, Aug-Sept 2016 and Jan-Feb 2017 maxima, but remarkably,
absent on SBUV, smaller on Dobson and Brewer and larger on GOME2, in consistence with the origin of
the seasonality described in section 5.


| INSTRUMENT | Bias | sdev | Seasonality |
|---|---|---|---|
| Dobson 62 | -0.7 | 0.4 | 1.5 |
| Brewer 40 | 1.6 | 0.3 | 1.2 |
| Dobson 62 (Ser) | -0.5 | 0.6 | 2.1 |
| Brewer 40 (Ser) | 0.3 | 0.3 | 0.9 |
| Dobson 62 (Ser + Temp Corr.) | 0.4 | 0.5 | 1.7 |
| SBUV | -0.3 | 0.2 | 0.9 |
| OMI-TOMS | -1.1 | 0.7 | 2.5 |
| OMI-DOAS | -1.2 | 0.8 | 2.5 |
| SUOMI-NPP | -0.4 | 0.6 | 2.2 |
| GOME2A | 1.3 | 0.7 | 2.5 |
| GOME2B | 2.1 | 1.1 | 3.9 |
| IASI-A | 1.7 | 0.6 | 2.1 |
| IASI-B | 1.9 | 0.5 | 2.3 |
| OMI Godfit v4 | 1.9 | 0.7 | 2 |
| GOME2A Godfit v4 | 1.7 | 1.1 | 3.3 |
| GOME2B Godfit v4 | 2.4 | 0.8 | 2.4 |

**Table 1** Per-cent bias, standard deviation and seasonality of the differences with Mini-SAOZ TOCs.
**6. Concluding remarks**
Mini-SAOZ Total Ozone Column (TOC) measurements carried out in Arosa between October 2015 and
March 2017 are fully consistent with Dobson and Brewer observations carried out during the same period
when corrected for ozone cross-sections differences and temperature dependences. The bias is smaller than
0.5% and the seasonality of the difference lesser than 1.7% amplitude. The seasonality is shown to be
coming from the Mini-SAOZ Air Mass Factors used to convert slant into vertical columns. On the satellites
side, the differences with ground based TOCs do show larger and higher biases variability among satellites,
as well as larger seasonality on all satellites, except on the Solar Backscatter Ultra Violet (SBUV)
instrument. Asides from the use of various ozone absorption cross-sections and wavelength ranges which
can induce biases of 1-2 %, most important for satellites TOCs differences with Mini-SAOZ is the larger
seasonality compared to GB direct sun measurements and far larger than with SBUV. This seasonality is
shown to be coming from the zonal mean profile climatology used for TOC calculations, ignoring
longitudinal and local variations. This applies to Mini-SAOZ zenith sky, as well to satellites nadir viewing
instruments, in contrast to limb profiler like SBUV or Dobson and Brewer direct sun observations, showing
smaller seasonality. In the future, if improved ozone profiles climatology is available, it may possibly even
reduce the 1-3% remaining seasonality between the various instruments.

**Acknowledgments** The authors thank Werner Siegrist and MeteoSwiss technicians for their help in Arosa
operations, Payerne data provision, WMO / GAW and the ESA/CCI (Climate Change Initiative Program)
and CNES for financial support. We thank C. Lerot and the ESA/CCI team for providing the GODFIT v4
overpasses above Arosa. We thank J. Hadji Lazarro and the IASI team for providing the IASI A and
IASI B overpasses above Arosa. We thank the French AERIS data infrastructure for providing access
to the ECMWF data used in this study.



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
