# Peer review of "Total Ozone Dobson, Brewer, Saoz and satellites"

_Atmospheric Measurement Techniques, 2019_

## Referee Comment (RC1) · Anonymous Referee #1 · 5 Aug 2019

The paper "Total Ozone Dobson, Brewer, SAOZ and satellites comparisons at the historical station Arosa" by J.-P. Pommereau et al. presents a study that compares total ozone measurements obtained from SAOZ instrument at Arosa with two other ground-based instruments (Brewer #40 and Dobson #62) at that location and a number of satellite overpasses over the station. The time period for the inter-comparisons is limited to about 1.5 year from October 2015 to March 2017. In the abstract and conclusions authors summarize results of the inter-comparisons and offer explanations for the observed discrepancies. Though, authors provide a very comprehensive analysis of ground-based comparisons, the analysis of results with satellite measurements is very shaky. Some of the conclusions are based on an erroneous assumption that SBUV is a

limb sensor, while it is a nadir sensor. I would suggest that authors either should collect more information regarding to different satellite retrieval techniques used in this study (perhaps, by contacting satellite groups), or rather show results of comparisons and avoid offering explanations for the observed differences. Another big concern to me is the statement that authors made in the abstract claiming that the agreement between SAOZ and Dobson/Brewer instruments is within 1% over 1.5 year and this agreement can provide "confidence to the long-term SAOZ measurements". This is a very short overlap that does not allow to judge the long-term stability of SAOZ. On my opinion, the manuscript requires a major revision, some facts need to be corrected, and the conclusions have to be re-evaluated. My specific comments are below.

Specific comments:

Title: On my opinion, the title of the manuscript does not accurately reflect the content of the study, since it focuses mostly on validation of SAOZ measurements. I would recommend authors to reconsider the title.

Page 1, Abstract, Line 19: I do not agree with the statement that agreement within 1% over 1.5 year can provide "confidence to the long-term SAOZ measurements". This is a very short overlap that doesn't allow to judge the long-term stability of SAOZ.

Page 2, Introduction, line 13: It would be nice if authors can explain what is "an irradiance default of linearity" and how it would affect ozone retrievals from SAOZ.

Page 4, lines 17-22: Authors demonstrate that corrections for O3 cross-sections improve comparisons. However, by looking at Figs.1 - 2 it remains unclear whether cross section corrections reduce the large noise seen in Fig.1 or the noise remain the same. Authors should add error bars on Fig. 2 or add a discussion about the noise into the text. The same argument goes to results shown on Fig. 3 due to temperature corrections.

Pages 6, line14: The instrument/satellite name is not "Nimbus-19" but "NOAA-19".

Page 6, lines 14-16: The name of the ozone sensor on board of Suomi NPP is OMPS. It is a joint NASA/NOAA mission. I suggest to change that phrase with "the NASA/NOAA Ozone Mapping and Profiling Suite (OMPS) on board of Suomi National Polar orbiting Partnership (NPP) satellite".

Page 7, Sections 4.1-4.2: Authors need to identify version of all data used in this study and provide references to the manuscript that describes the algorithm or validates the data. It is not enough to provide a link to the web page as a version of data can change.

Page 7, Section 4.3: This section provides incorrect information about SBUV. First, SBUV is a nadir sensor not a limb sensor as it noted in this section. Secondly, it has 12 spectral channels not 4. The name of SBUV/2 sensor used in this study over the 2015-2017 time period is "NOAA-19" not "Nimbus-19". A proper reference to the SBUV v8.6 algorithm is (Bhartia et al., 2013). I didn't find (Bhartia et al., 1996), referred in line 22, in the 'References'.

Page 7, Section 4.4, Title: All sections are named after sensor's names, not satellite's names. Should then the section title be "OMPS"?

Page 7, line 35: Incorrect reference to the TOMS algorithm. This is one of references to TOMS v8 algorithm: Bhartia and Wellemeyer, 2002:

Bhartia, P. K., and C. W. Wellemeyer, "OMI TOMS-V8 Total O3 Algorithm", Algorithm Theoretical Baseline Document: OMI Ozone Products, P. K. Bhartia (ed.), vol. II, ATBD-OMI- 02, version 2.0, Aug. 2002.

Page 8, lines 1-2: Suomi NPP OMPS is on a polar orbit with the equatorial crossing time 1:30 pm or 13:30. Sensor can make measurements of the sunlit part of the Earth, therefore most of measurements are collected in ascending mode.

Page 8, lines 13-16. Authors need to clarify what "OMI-D" or "OMI-T" stands for. I assume "OMI-T" stands for OMI data processed with NASA TOMS algorithm, and "OMI-D" for OMI data processed with KNMI DOAS algorithm. But this should be clarified in

the text.

Page 8, line 17: What does "the frequent masking of satellites" means? Do you mean "cloud mask" or some quality flags?

Page 8, line 24: All instruments here are listed by their names except for OMPS that is listed as "NPP".

Page 8, lines 25-28: This is correct that SBUV v8.6 algorithm retrieves ozone profiles, and the total ozone is calculate as a sum of all layers. But SBUV is a nadir sensor.

Page 8, lines 29-31. This sentence is very confusing. It's not clear which instrument uses "Averaging Kernels derived from TOMS v8 zonal mean climatology". This part of the text needs major revision. I would recommend authors to find more information about different satellite algorithms to provide comprehensive explanations. Alternatively, they can show results of comparisons but avoid explanations for the observed differences.

Page 11, lines 15-17: Again SBUV is a nadir sensor. SBUV algorithm also uses seasonal zonal mean climatological profiles as the a priori (see Bhartia et al., 2013).

Technical comments:

Pages 1-2, Abstract/Introduction: There are number of abbreviations that are used for the first time in the manuscript (like SAOZ, OMI, SBUV, CINDI, CCD etc.) that are not explained.

Page 8, line 24: should be comma "TOMS v8, SBUV".

Page 8, line 28: What does "GB" stands for? Ground-based?

Page 11, line 9: It seems that "larger" and "higher" mean the same here.

Page 11, line 13: What does "GB" means?

---

## Referee Comment (RC2) · Anonymous Referee #2 · 11 Sep 2019

The manuscript "Total Ozone Dobson, Brewer, Saoz and satellites comparisons at the historical station Arosa" by Pommereau et al. describes the comparisons between different ground-based total column ozone measuring instruments (Dobson, Brewer and SAOZ) at Arosa station during the period October 2015 to March 2017. Additionally, the SAOZ measurements are compared to total column ozone measurements from a suite of satellite instruments. The measurement differences between the individual instruments are discussed and mostly explained, however, some parts of the manuscript describing the methods are very brief and, in my opinion, incomplete. Stable ground-based measurements are very important for satellite validation, so the overall topic of the manuscript is interesting and valuable. While the comparison of the ground-based

instruments is mainly well described, I thought that the description of the satellite-SAOZ comparison was too brief and lacking some important details. Therefore I recommend the manuscript for publication with major revisions, and I would like to see the following comments and suggestions considered.

**General suggestions/comments:**

- As mentioned in the previous paragraph, in my opinion, Section 5 of the manuscript is lacking detail in the description of the applied methods, and also in the discussion of the results. It is not clear what temporal resolution the compared measurements have (monthly means are mentioned in the beginning of the section, however Figure 4 shows the unit of the x-axis as "Day of Year"). Additionally, it is not clear how the individual measurements (obtained multiple times a day, once a day, every other day?) were aggregated to the displayed time unit. The aggregation could make a difference when comparing the individual measurement systems. It is also not explained what kind of spatial aggregation has been performed. Were the point measurements of the SAOZ (with the coordinates of Arosa) compared to a whole satellite measurement grid cell? This also could cause differences in the comparisons. I highly recommend that Section 5 is reworked thoroughly, and a lot more information about the methods, and also discussion of the results is added.

- The description of the results in Section 5 seems a little unstructured and maybe not well formulated. I recommend rephrasing this section with great caution to make sure that descriptions are correct and understandable.

- How exactly is the overall bias between two instruments calculated? Table 1 only provides one value for the bias, however, in all figures the bias between instruments is not static at this value, but changes over the displayed period. Please describe the underlying method here.

- Some sentence structures are too complicated and not grammatically correct. I recommend having a native speaker look over the manuscript to identify and correct these problems.

**Minor comments:**

- Page 1, line 1: Should "Saoz" not be "SAOZ" (all capitalized)?

- Page 1, line 8: Remove the "the" after "importance of"

- Page 1, line 10: Should be "instrument" not "instruments"

- Page 1, line 12: Should be "data set" not "data sets"

- Page 1, line 13, 14, 17: Should there be references in the abstract?

- Page 1, line 14: Is there the word "instrument" missing after "OMI AURA"?

- Page 1, line 16: Should be "satellite" not "satellites"

- Page 1, line 25: I would remove the word "remarkably" since it is not clear why this is remarkable

- Page 1, line 27: The word should be "Aside" not "Asides". There are several of these throughout the manuscript that should be corrected.

- Page 1, line 27: Should be "satellite" not "satellites"

- Page 1, line 28: Should be "satellite measurement" not "satellites measurements"

- Page 1, line 29: "are" missing between "satellites" and "sensitive"?

- Page 2, line 6: Explain "CINDI".

- Page 2, line 33: I think a word like "measured by", "obtained by", or "provided by" is missing between "are" and "the"

- Page 4, line 9: The minimum of the comparison Brewer-SAOZ in Figure 1 is not in March, but in May. Maybe rephrase this part of the sentence?

- Page 4, Figure 1: Why does the Brewer data only start at the beginning of 2016?

- Page 4, Figure 1: Why are there no error bars on the Brewer-SAOZ comparison?

- Page 5, lines 7-9: The meaning of the sentence starting with "However, while…" is not clear. Please rephrase.

- Page 5, lines 21-25: This is one sentence. It is too long and therefore really hard to understand. Please rephrase. Also, this paragraph describes the explanation for the differences, and for that it is actually too short, in my opinion. Please add more detail here.

- Page 5, line 25: Is there a reference for the seasonal differences in Payerne ozone-sonde peak ozone levels?

- Page 6, lines 8-9: This reference to the top part of Figure 1 is not clear. The differences between ground-based measurement systems and SOAZ are shown in the bottom panel of Figure 1, so should the reference not be for that part of the figure?

- Page 6, line 18: IASI was defined in line 17 without the addition of "A" and "B". So it is confusing to mention these two IASI options here. Please add their definition to line 17.

- Page 7, line 27: Shouldn't the header of this section be "OMPS" instead of "SUOMI-NPP"? All the other section headers are named after satellite instruments not satellites.

- Page 9, lines 9-10: This sentence does not seem correct in its meaning: radio-sonde profiles do not provide ozone profiles.

- Page 9, line 18: The bias scatter is reduced, but the bias is still high with +1.7% to +2.4%!

- Page 10, Figure 5: The lines for GOME2A, GOME2B and OMI are shorter than the Brewer and Dobson lines, why?

- Page 11, lines 14-15: It is mentioned here for the first time that the ozone climatology used for the TOC calculations is actually a zonal mean climatology, and that the longitudinal and local variations could be causing some of the differences. I think this is a piece of information that should be mentioned earlier already!